# Mineralogical and Physico-Chemical Characterization of the Oraşu-Nou (Romania) Bentonite Resources

**Gheorghe Damian [1],\*, Floarea Damian [2], Zsolt Szakács [2], Gheorghe Iepure [2] and Dan Aştefanei [1]**

1   Department of Geology, Faculty of Geography and Geology, Alexandru Ioan Cuza University, 700505 Iaşi, Romania; astdan@uaic.ro

2   North University Center of Baia Mare, Technical University of Cluj Napoca, 430083 Baia Mare, Romania; loricadamian@cunbm.utcluj.ro (F.D.); zsolt.szakacs@cunbm.utcluj.ro (Z.S.); iepureg@cunbm.utcluj.ro (G.I.)

\*   Correspondence: gdamian_geo@yahoo.com

**Abstract:** The objective of this study is to describe the mineralogical composition and chemical properties of the Oraşu Nou bentonite, from northwestern Romania. For mineralogical determinations, the following were used: X-ray diffraction (XRD), Fourier transform infrared spectroscopy (FR-IR), thermogravimetric analysis, scanning electron microscopy coupled with energy-dispersive X-ray spectroscopy (SEM-EDX). The chemical compositions and physical properties of the bentonites and bentonitized rocks were also determined. Calcium type montmorillonite is the predominant mineral in this deposit. Its average mass fraction is between 35% and 75%, reaching up to 95%. A small amount of halloysite and very fine cristobalite were also identified in the fine fraction. Quartz, feldspar, and kaolinite were identified as impurities. The average pH of natural bentonite is 6.2. Its cation exchange capacity (CEC) is in the lower-middle range for smectites at 45.89 cmol/kg, absorption capacity 43.58 mL/g, swelling degree 9.41%. Because of the high amounts and purity of montmorillonite, the valuable component mineral, the way is open to an easy refinement of this important resource. This way very high-quality colloidal suspensions can be obtained which can be used in the most modern applications of micro- and nanostructured materials.

**Keywords:** bentonite deposit; montmorillonite; cristobalite; halloysite; nano- and micro-structured materials; physico-chemical properties





## 1. Introduction

Bentonites usually contain clay minerals (predominantly montmorillonite and secondarily kaolinite and illite), cristobalite, carbonates, zeolites, iron oxides and hydroxides, and relics of quartz and feldspar. They are widely used in their native forms, or after refinement, purification, or activation in many areas, because of high absorption rates displayed towards water or organic compounds, high cation exchange capacity and protein sedimentation properties [1–4]. The name bentonite was proposed for the first time for the cretaceous clays found in the formations from Benton, Wyoming, USA by Knight [5]. Montmorillonite, its principal mineralogical component, was discovered in the type locality of Montmorillon, Vienne, Poitou-Charentes, France.

The main mineral component of bentonites is montmorillonite, a member of the smectite group, a hydrated di-octahedral alumino-silicate. Its general formula can be written as $(Na,Ca)_{0.33}(Al,Mg)_2(Si_4O_{10})(OH)_2 \cdot nH_2O$. In contact with water, it can reversibly swell up to 15 times of its original volume, forming gelatinous colloidal suspensions or plastic films [6]. Subordinate minerals can be cristobalite, tridymite, feldspar, halloysite, kaolinite [7,8]. The various forms of silica present in the bentonite deposits can be opal, cristobalite and quartz. Opaline silica is not considered to be harmful to humans, in contrast to cristobalite [9].

In Romania, bentonitic clays appear in large quantities, with important deposits in the Transylvanian Basin (Răzoare, Valea Chioarului, Oraşu Nou, Ocna Mureş, Ciugud, Sănduleşti-Petreşti, Gurasada), or in western Romania's Banat region (Brebu, Breaza), [10].

Important bentonite clay mineral resources are available around Oraşu Nou in NE Romania [11]. This deposit is situated at 30 km from the municipal cities of Baia Mare and Satu Mare, at the entrance of the Oaș region's plateau. This area was explored with drillings revealing high quantities of bentonitic rocks. The bentonite bearing geological formations were studied 50 years ago [12]. Some quarries were opened but nowadays the exploitation slowed to a halt, because of non-sustainable results. In this whole period, there was no complete description of physical, chemical, and structural properties of the minerals contained in the bentonite deposit, using modern physico-chemical methods. Therefore, no evaluation of their utility in a high-value refined, purified, or activated form exists, with modern nano- and microscale applications. Bentonite consists mainly of montmorillonite, with cristobalite as subordinate mineral, along with small amounts of quartz, feldspar, halloysite, kaolinite [11]. From a mineralogical point of view, chemical alterations transformed the volcanic rocks in clay minerals, while the montmorillonite was the prevailing one [13]. This mineral has formed through the break-down of feldspars, of the volcanic ash, and of the volcanic glass from the perlites.

Similar bentonite resources can be found in the Carpathian area such as those from the Hliník nad Hronom deposit in Slovakia [7], but also in the world in Spain and in Turkey [8,14].

Bentonites in various refinement stages are used in over 30 industrial branches or in environmental protection [3,15,16]. Its pharmaceutical applications are well known, for example under the commercial name "Smecta", as an antidiarrheic, anti-flatulent, bowel protective against viruses and bacteria, and as a detoxifying agent [10].

The resources of this region represent an important bentonite source. The possibility of open-pit extraction represents another important advantage. The properties and industrial applications of bentonites are a consequence of their mineralogy. The CEC capacity and the high plasticity index give them the possibility to be used in many fields [17].

The main part of bentonites extracted on a world scale is used in foundries and as drilling-mud [18]. Smaller quantities are used in special applications. According to Kuzvart [19] and Patterson and Murray [20], bentonites are used as ingredients in ceramics, in waterproofing or as sealant in civilian construction; petrochemistry, as a catalyst in raw petrol refinement and during the filtration of the products obtained after fractional distillation; to obtain white or Portland cement.

Meunier [16] recommends using bentonites in the paper industry to obtain white glossy paper; in pharmaceutical and cosmetics industry in obtaining liniments, wet compresses, soaps, make-ups and powders, creams; in the food industry for the purification or as a rinsing agent for water, wine, beer, vegetable-oils.

In agriculture, it enhances sandy soils, helping water retention, and, after drying-out, soil aeration; or it can be used as a feed supplement for animals [21,22]. Plant nutrition is also sustained by intermediation of base ion exchange, gradual fertilizer desorption and retention of pesticides [23,24]. Bentonites with a high montmorillonite content similar to the one found in Oraşu Nou can be used to immobilize fungicides or insecticides [25]. Application of Ca-bentonite from Oraşu Nou as a soil amendment, can reduce soil water losses, significantly increase microbial biomass parameters, and crop yield [26]. Its capacity to absorb pesticides was applied to detoxification of paraquat poisoning victims [27].

The use of bentonites in environmental protection is reduced when compared to that of natural zeolites. Bentonites mixed with zeolites have been used to inactivate heavy metals in polluted soils, especially zinc [28]. They can also be used in pollutant removal processes for industrial or civilian residual waters [29,30], for the purification of industrial gases, protection of waste storage areas [31], including nuclear waste storages [32,33]. Bentonites from the Oraşul Nou deposit can be considered a potential buffer material for isolation of radioactive waste, where they can be used due to their low permeability and high CEC [34].

Bentonite blocks serve as a sealing material around waste canisters to isolate them from the surrounding atmosphere [35]. Recent research applied bentonite-based materials for the absorption of ammonium or heavy metal ions from wastewaters, especially for the retention of chromium [36].

## 2. Geological Setting

The geological formations from the Orașu Nou region (Figure 1a) are of Neogene age and they consist of sedimentary and volcanic deposits [12]. The volcanic formations are lava flows intercalated in sedimentary formations, that Sagatovici [37] and Bârlea [12] identified as being of Badenian age. Later, Rădan et al. [38] established that they were of Pannonian age.

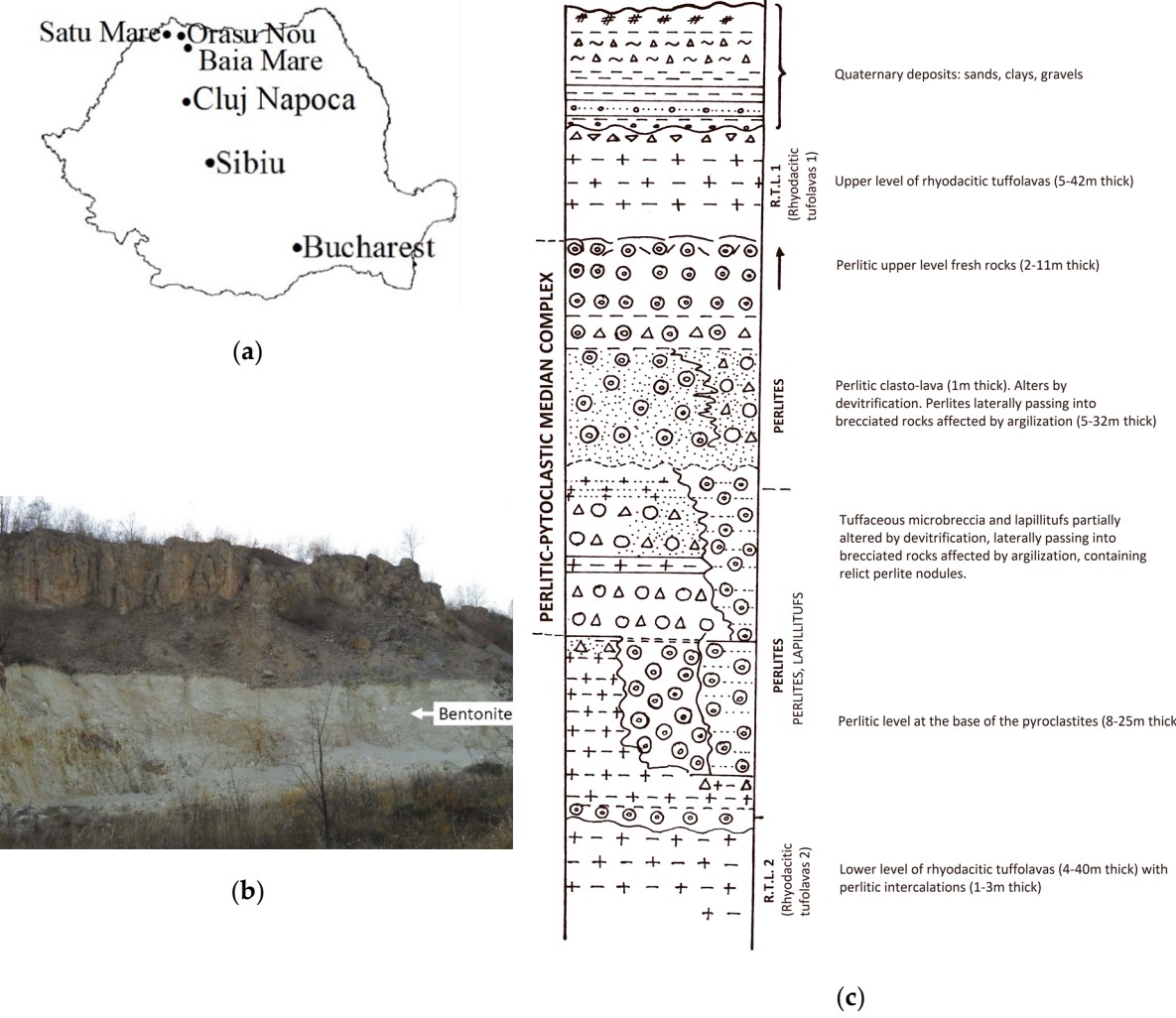

**Figure 1.** General view of the bentonite deposit from Orașu Nou. (**a**) Location of the bentonite deposit; (**b**) The bentonite layer exposed in one of the sampling quarries; (**c**) Synthetic litho-stratigraphic column representation of the geological formations.

The bentonite resources are hosted in volcanic formation. In Figure 1b. the volcanic rock deposits, with thicknesses up to 300 m, are represented by ignimbrite, perlite-facies rhyodacites and pyroclastic rocks from the Pannonian age. The bentonite deposits resulted from the alteration of the pyroclastic and perlite layers by hydrothermal-deuteric alteration.

The litho-stratigraphic column representation in Figure 1.c shows that rhyodacite and tuffo-lava layers appear in ignimbritic facies in the lower (R.T.L. 2) and upper parts (R.T.L. 1). The median complex level is perlitic, consisting of lapilli and cinerite tuffs intercalated between two layers of rhyodacite perlites. The pyroclastic rock fragments are

represented by hyaline rocks, volcanic rocks, metamorphic rocks, marls, and siltstones. The matrix of pyroclastic rocks is made of cinerite material. The rhyodacite perlites contain up to 95% volcanic glass as their main constituent and plagioclase felspars, sanidine, quartz and biotite. The median complex is pervasively substituted with montmorillonite and cristobalite forming the bentonite rocks, as can be seen in the outcrop in Figure 1b. The deposits have lenticular shape 100–400 m long, 50–250 m wide, and 3–15 m thick. In these deposits, the bentonite rocks have a white color, a compact aspect and mainly a perlite structure.

## 3. Materials and Methods

### 3.1. Materials

To obtain a more complete picture, we collected multiple samples of white bentonite material, the more valuable type, along with partially bentonitized tuffs of lower economic value. The study material was collected from the northern part of the Orașu Nou area from the bentonitized Pannonian volcanic formations (Figure 2). The position of the selected representative raw bentonite samples can be seen in Figure 2 and the sampling point coordinates in Table 1.

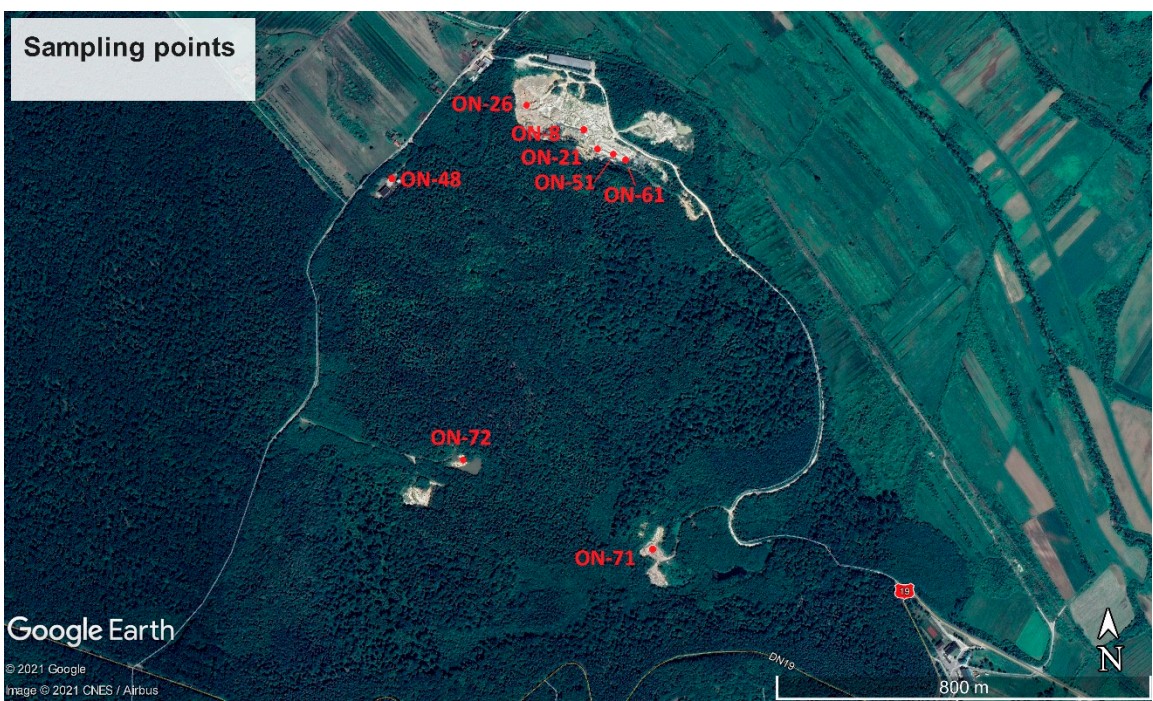

**Figure 2.** Position of the sampling points Orașu Nou area represented using a Google Maps base map [39].

**Table 1.** Location and details of the representative bentonite and bentonitized rock samples.

| No. | No-Sample | The Type of Rock | Latitude | Longitude |
|-----|-----------|------------------|----------|-----------|
| 1 | ON-48 | White bentonite | 47.85441862 | 23.25203075 |
| 2 | ON-26 | White bentonite | 47.85582356 | 23.25591421 |
| 3 | ON-8 | White bentonite | 47.85528428 | 23.25737839 |
| 4 | ON-21 | Perlite bentonitized | 47.85493308 | 23.25781122 |
| 5 | ON-51 | Lapilli tuffs bentonitized with cristobalite | 47.85482907 | 23.25829366 |
| 6 | ON-61 | Tuffaceous microbreccia with halloysite | 47.85472446 | 23.25862237 |
| 7 | ON-71 | Perlite bentonitized | 47.84740083 | 23.25927758 |
| 8 | ON-72 | Rhyodacitic tuffolavas | 47.84911822 | 23.254001468 |

These samples were processed in parallel determining the maximum and minimum values of the different chemical characteristics and mineral contents in this deposit. We dedicated the main part of our study to the white bentonites and the nano- and microstructured materials that can be easily obtained from them, because they are of the greatest economic and scientific interest. A basis of comparison can be found in other publications [40,41]. A representative picture of selected white bentonite samples used in our determinations can be seen in Figure 3.

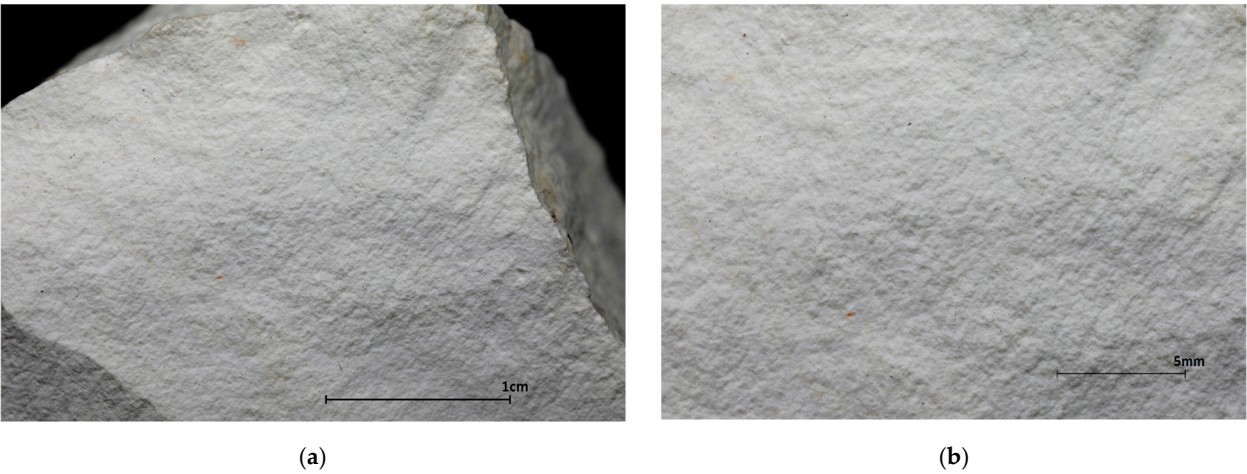

(**a**)                                                                                           (**b**)

**Figure 3.** Representative image of the white bentonite sample from Orașu Nou used in the determinations. (**a**) Bentonite fragment; (**b**) Detail in the bentonite sample.

### 3.2. Methods

For the granulometric study each sample was mixed with distilled water until a suspension was obtained. The separation of the granulometric fractions was made by successive wet sieving and finally by sedimentation. The sieves had the following mesh sizes: 0.5, 0.25, 0.16, 0.10, 0.08, 0.063, 0.04 and 0.01 mm.

To further separate the clay fraction obtained from the white bentonite, the suspension obtained after the last sieving was split in volumes of 0.75 L. Each of these fluids was stirred for 30 min, using a stainless-steel propeller-blade stirrer. To sample the sedimented fractions, Kubiena pipettes were used. To obtain the nano-and micro-structured material of the greatest interest, the supernatant was sedimented for 24 h and the remaining liquid was collected. Water was evaporated from this fluid and the leftovers were dried-out at room temperature. The first couple of millimeters were scraped away from the obtained shiny surface using a celluloid film. It was manually powdered using an agate mortar and pestle, until it passed through a 10 μm sieve.

To describe the bulk material, we used parts of the representative samples that we collected and X-ray diffraction [42]. This was done on a previously calibrated Shimadzu XRD-6100, (Shimadzu Corporation, Kyoto, Japan) (filtered Cu-K$_{\alpha 1}$, U = 40 kV, I = 30 mA, 0.5°/min scan speed), followed, when needed, by a quantitative Rietveld phase analysis and refinement. Thermal analysis was done using a MOM-Derivatograph (MOM Kft., Budapest, Hungary) with TGA-DTA at 10 °C·min$^{-1}$ heating rate, from ambient temperature to 1000 °C. For a correct identification of the silica polymorph found in the bentonite samples, XRD and thermogravimetric analyses (TGA) were necessary [43].

The FT-IR transmission spectrum was measured using the KBr-pellet method on a Bruker Vertex 70V (BRUKER Optic, GmbH, Ettlingen, Germany) (4 cm$^{-1}$ resolution, 380–4400 cm$^{-1}$ domain). SEM imaging and EDX analyses were made with Zeiss Merlin scanning electron microscope (SEM) (ZEISS, Jena, Germany), with a Gemini II column and Schottky type electronic source, at the Geological Institute of Romania. EDX analyses were performed at an acceleration voltage of 15 kV, with an electron beam intensity of 0.7–1.5 nA, using the Oxford Instruments X-MAX 50 Si EDXS (Oxford Instruments, Abingdon, UK)

detector at $-60\ ^\circ$C, attached to the microscope. SEM imaging techniques were applied only to the raw white bentonite. All analyses were preceded by an overnight desiccation at $110\ ^\circ$C [2].

Whole rock analysis was made by fused bead, acid digestion and ICP-AES analysis (PerkinElmer Inc., Waltham, MA, USA). Loss on ignition (LOI) was determined by furnace or TGA. The pH meter was a Multiparameter type Multi 340i (WTW, Weilheim, Germany) with a Sentix electrode calibrated at pH = 7 by a STP7 standard solution. Determination of Cation exchange capacity (CEC) was made using the ammonium acetate saturation method [2,44]. The bentonite samples were saturated with $NH_4^+$ using 1 M concentration $CH_3COONH_4$ at a pH of 7.0. Quantitative CEC determinations were made using replacing solutions. Atomic absorption spectrophotometry was used for Ca and Mg, while flame spectrophotometry for Na and K. The specific gravities of the samples were determined using the pycnometer method. Absorption capacity is the property of retaining large amounts of water, with an increase in volume 12–15 times greater than that of the dry. Plasticity index was determined by the rolling method, according to ISO-17892-12:2018.

## 4. Results and Discussions

### 4.1. Description of the Raw Bentonitized Samples

Bentonite rocks form lenticular bodies up to 15 m thick and up to 400 m long. In the Orașu Nou deposit the bentonite is a rock with a massive compact aspect, sometimes friable, white-colored with slight yellow or gray nuances, indicating a high-quality material (Figures 2 and 3). White bentonites with a grainy structure have high cristobalite contents.

The results of the conducted granulometric studies on every raw sample are presented in Table 2. The clay fraction is predominant in the white bentonites, while the fraction above 0.5 mm is characteristic for the poorly bentonitized material. These results, corroborated with the presented geological studies, can indicate the expected high contents of clay minerals in the bentonites and the presence of primary mineral relics (quartz, feldspars and fragments of fresh rocks) in the partially bentonitized samples.

**Table 2.** Maximum and minimum mass percentages of the granulometric fractions of bentonites and bentonitized rocks.

| Granulometry | (%) | |
| --- | --- | --- |
| | Maximum | Minimum |
| +0.500 mm | 58.10 | 1.30 |
| +0.250 mm | 31.30 | 0.50 |
| +0.160 mm | 11.45 | 2.70 |
| +0.100 mm | 10.69 | 3.30 |
| +0.080 mm | 9.70 | 4.40 |
| +0.063 mm | 8.35 | 1.53 |
| +0.040 mm | 6.22 | 1.22 |
| +0.010 mm | 35.50 | 4.89 |
| −0.010 mm | 35.10 | 5.00 |

The results of the conducted XRD studies, regarding the mineralogical composition of the envisaged samples, are summarized in Table 3. The distribution of montmorillonite in bentonite is variable. Montmorillonite is mainly concentrated in the finest fractions.

As could be expected, the XRD results underline the general geological observations. Quartz, plagioclase feldspars (like albite, oligoclase) and sanidine participate in the composition. These minerals can be mainly found in the poorly bentonitized rocks, or in the granulometric fractions above 0.25 mm.

The main mineral component formed by alteration is montmorillonite. Its average content is ranging from 60–65% up to 95% in the different bentonite samples. Cristobalite, kaolinite, and, in lower mass percentages, tridymite, and halloysite were identified alongside. Iron oxyhydroxides and some carbonates are also present.

**Table 3.** Synthetic table of the minerals and their mass fraction limits determined using XRD in the studied bentonites and bentonitized rocks. (Sp = sporadically).

| Mineralogical Composition | Bentonite | | Bentonitized Rock Samples | |
|---|---|---|---|---|
| | **Max.** | **Min.** | **Max.** | **Min.** |
| Montmorillonite | 95 | 60–65 | 55–60 | 12–20 |
| Cristobalite | 30 | 3–4 | 60 | 30 |
| Plagioclase feldspar (Anorthite 35–50%) | 4 | | 4–5 | 2 |
| Quartz | 3–4 | 1 | 6 | 0.5 |
| Sanidine | 3 | | 3–4 | 1.5 |
| Tridymite | 5–7 | | 7–9 | 1–2 |
| Kaolinite | 8–10 | 1 | 8–10 | |
| Illite | 2–3 | 1 | 3–4 | 1 |
| Halloysite | Sp | Sp | Sp | Sp |
| Pyrite + marcasite | 1–1.5 | 1 | 2–3 | 1 |
| Iron oxyhydroxides | 2–3 | 0.5 | 3–4 | 1 |
| Carbonates | 3–4 | | 4–5 | 0.5 |

## 4.2. SEM-Imaging

The characterization of these minerals needs a complex methodology to be applied. Identical mineralogical compositions were also reported for the replacement standard montmorillonite STx-1b or other similar samples studied before [41].

A suggestive SEM micrograph of a freshly broken chip from the raw white bentonite sample can be seen in Figure 4. It can immediately give a general idea of the mineralogical composition that should be expected. This image can also be compared to those from other publications [40,45,46].

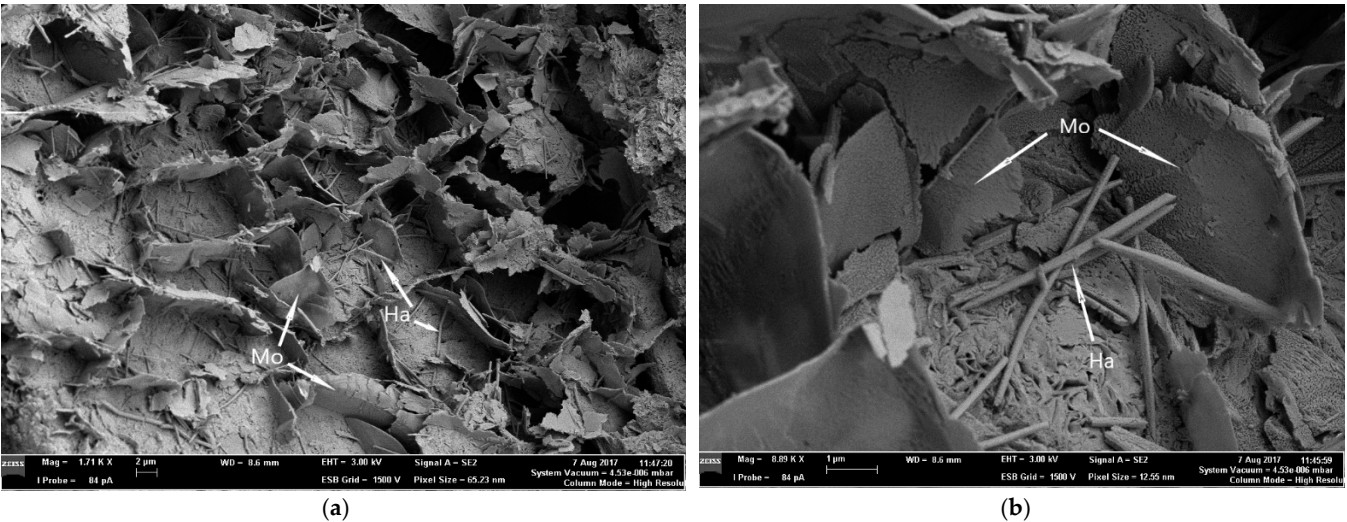

(**a**)    (**b**)

**Figure 4.** SEM micrograph of a freshly broken sample of bentonite from the white Orașu Nou; (**a**) massive montmorillonite flakes (Mo) and halloysite nanotubes (Ha); (**b**) detail with montmorillonite flakes (Mo) and halloysite nanotubes (Ha).

The presence of montmorillonite is proven by the typical structures which are at least 5–7 μm long and a few tens of nanometers thick. SEM images indicate high crystallinity of montmorillonite. Scanning electron microscope images also showed that montmorillonite is abundant in the Orașu Nou bentonite samples. Montmorillonite flakes (up 7 μm in diameter) occurred as massive flakes [14].

The presence of halloysite is suggested by the rod-like mineral structures, with average lengths of 5μm, and average diameter of 175 nm, shape which is characteristic for this mineral (Figure 4). Halloysite appears as impurity in the bentonite raw material.

Microscopically it has also been identified as small nests associated with cristobalites in partially bentonitized rocks.

### 4.3. X-ray Diffraction Studies

Because in the case of montmorillonite, as for the other clay minerals, the best identification can be made using XRD, this study must be discussed first. The obtained diffractogram of the refined nano-/microscale separate can be seen in Figure 5.

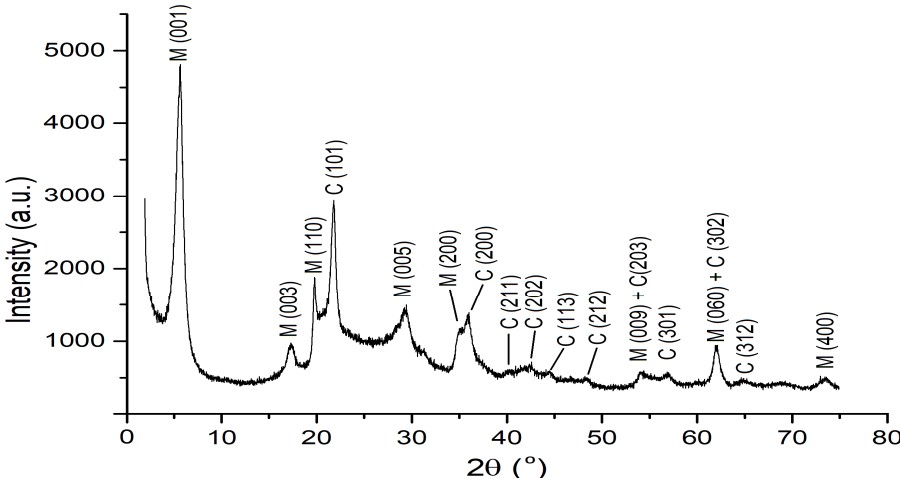

**Figure 5.** The XRD pattern of the white bentonite sample separate from Orașu Nou (M = Montmorillonite, C = Cristobalite).

As for other clay minerals, the most important identification criteria is the position of the basal peak, which should be in the domain d(001) = 15.45–15.938Å. This value is highly influenced by the interchangeable cation's type and the number of absorbed molecules, which, in our case, were only water remaining after desiccation [47]. As can be seen in Table 4, this value indicates a Ca-montmorillonite. The high value for the peak obtained from the anionic complex is equilibrated by the penetrated interchangeable cations $Ca^{2+}$, $Fe^{2+}$, which also leads to high water absorption properties. However, as seen in the diffractogram, the very intense and sharp peak indicates a well crystallized montmorillonite, whose structure is not disturbed by interlamellar cations or water molecules. The shape of the peaks also proves the lack of montmorillonite-illite interstratification.

**Table 4.** Synthetic table of the minerals and their mass fraction limits determined using XRD in the studied bentonites and bentonitized rocks (A.U.—arbitrary units; vs—very strong, s—strong, m—medium, w—weak).

| Montmorillonite Type hkl | ASTM 13-135 [48] | | Na-Montmorillonite [49] | | Montmorillonite Orașu Nou | |
|---|---|---|---|---|---|---|
| | d (Å) | I (%) | d (Å) | I (A.U.) | d (Å) | I (A.U.) |
| (001) | 15.0 | 100 | 12.63 | 237 | 15.451 | vs |
| (003) | 5.01 | 60 | | | 5.018 | m |
| (110) | 4.50 | 30 | 4.48 | 160 | 4.66 | s |
| (004) | 3.77 | 30 | 4.389 | 47 | | |
| (005) | 3.02 | 60 | 3.136 | 47 | 3.021 | s |
| (200) | 2.58 | 40 | 2.564 | 23 | 2.551 | s |
| (006) | 2.50 | | 2.49 | 76 | 2.493 | s |
| (007) | 2.15 | 10 | 2.014 | 54 | 2.129 | s |
| (008) | 1.88 | 10 | | | 1.879 | w |
| (009) | 1.70 | 30 | 1.67 | 4 | 1.695 | w |
| (060) | 1.50 | 50 | 1.50 | 26 | 1.495 | s |

The (003) peak was identified in all diffractograms to have a medium intensity and slightly elongated shape. The (110) peak is also sharp and intense, being mainly characteristic to Na-montmorillonite. The other peaks (005), (008), (009) indicate Ca-montmorillonite, and have reticular distance values close to those specified by the ASTM 13-135 file [48]. The (200) peak is superimposed over the cristobalite peak, so it seems relatively weak and bell-shaped, but it can be identified in all our measurements. The relatively intense (060) peak indicates the presence of relatively high quantities of Fe in the octahedral positions, which also leads to an intensification of the (001) peak.

A Rietveld refinement and analysis of the components present in this nano-/microscale montmorillonite separate shows a composition of 66% montmorillonite and 33% cristobalite. Quartz or feldspars were not identified in our separate, contrary to the raw samples and the ones from other studies [40].

The heat treatments were performed by burning the samples at 550 °C for one hour, after which they were analyzed by XRD. Górniak et al. [7], Nemecz [49] and Thorez [50] show that by burning montmorillonite at 550 °C the peak at d(001) disappears from d = 14–15 Å and moves to about 10 Å. In the heat-treated samples (Figure 6a) we observed the disappearance of the peak at d(001) = 15.93 Å and the appearance at d(001) = 9.73 Å with a high intensity decrease. This modification also confirms that the analyzed sample corresponds to montmorillonite.

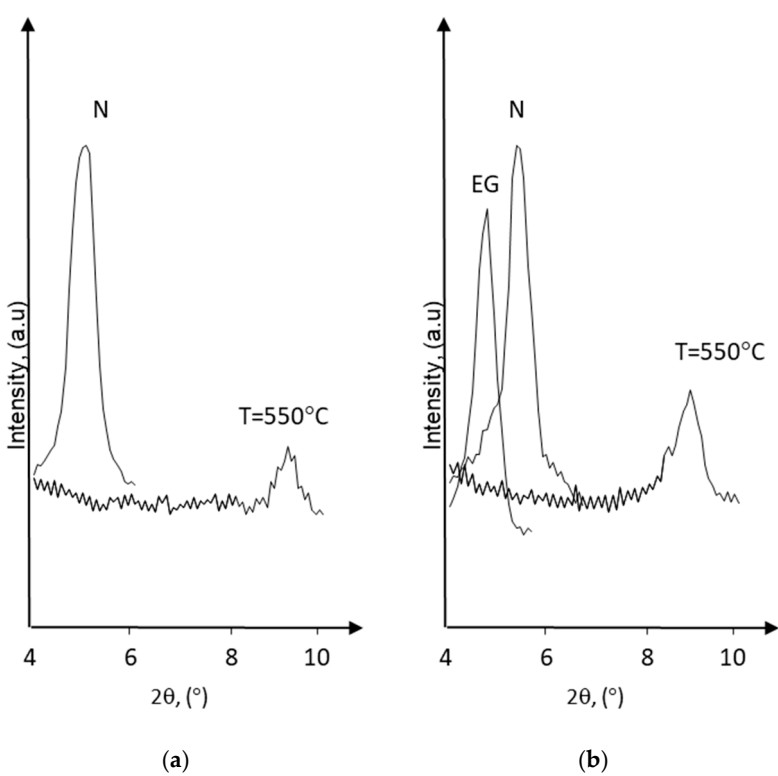

**Figure 6.** XRD patterns of the bentonite from Oraşul Nou: (**a**) comparison of the bulk (N) and calcinated sample, (**b**) oriented and ethylene glycol saturated (EG) sample compared to bulk (N) and calcinated samples.

Ethylene glycol treatment was also performed. The suspension with montmorillonite particles was sedimented on glass slides. The oriented sample was treated in a closed atmosphere saturated with ethylene glycol vapor for 6 h after which it was analyzed by XRD (Figure 6b). By glycolation, the value of reflex d(001) should increase from 14–15 Å to 17 Å [28,49,50]. The XRD spectra of ethylene glycol saturated-oriented sample from Figure 6b shows the displacement of montmorillonite's the principal peak (001) from 15.93 Å to 17 Å. This proves the exclusive presence of expanding layers of montmorillonite [7,49–51].

Cristobalite is another notable mineral component of the bentonites and bentonitized rock from Orașu Nou. Generally, it is deposited from high alkalinity colloidal solutions. It is present in all rocks independently of their bentonitic alteration state, but mainly in the coarser fractions. However, it can be seen from the above studies, its presence is also important in the fine, nano-/microscale fractions. Increasing quantities of α-cristobalite are proportional with the degree of bentonitic alteration. Besides the diffractogram obtained from the nano-/microscale material from Figure 5, another diffractogram, showing large quantities of α-cristobalite in the raw sample, can be seen in Figure 7.

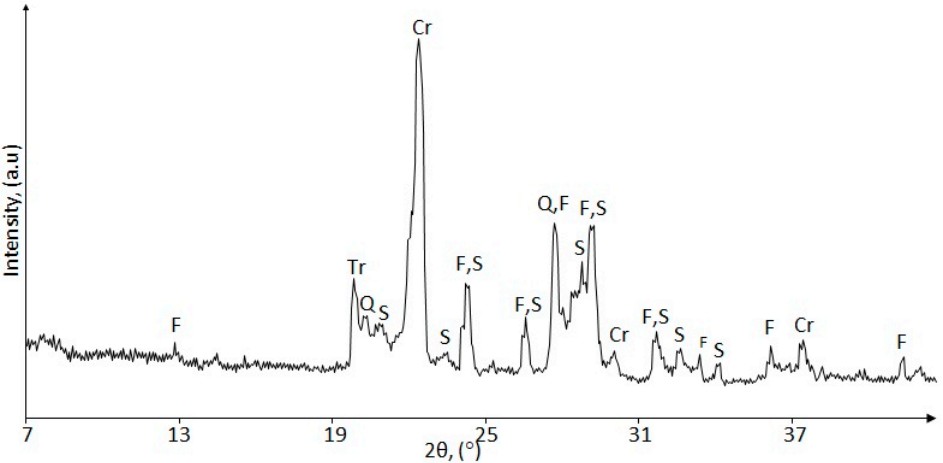

**Figure 7.** Sample XRD pattern containing high quantities of cristobalite (Cr—Cristobalite; Tr—tridymite; Q—quartz; S—sanidine; F—Plagioclase Feldspar).

The diffraction peaks can be attributed to the α-cristobalite according to the ASTM 11-695 file [48]. In our samples, its presence is mainly characteristic to the coarser fractions. The most intense diffraction peaks are obtained at d = 4.07 Å, alongside the ones at d = 3.139 Å, d = 2.855 Å, d = 2.492 Å, indicating this low temperature, rapidly cooled variety of silica. The discussed diffractogram also exhibits some quartz which is still not yet devitrified, and tridymite, another alteration product of quartz and polymorph of silica. Of course, the primary plagioclase felspars can also be identified.

X-ray diffraction was used for the identification of halloysite. A typical sample diffractogram, containing larger quantities of this mineral, can be seen in Figure 8.

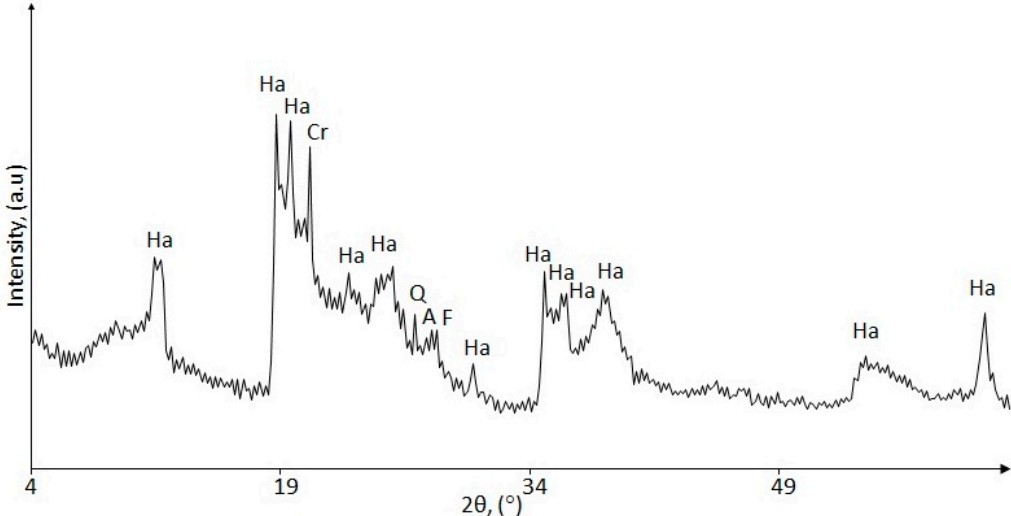

**Figure 8.** Sample XRD pattern containing high quantities of halloysite (Ha—Halloysite; Cr—Cristobalite; Q—Quartz).

The low-intensity lines indicate a mineral with poor crystallinity. However, identical diffractograms were previously obtained for dehydrated halloysite [48] or for bleached halloysite from Dunino mine in Poland by Szczepanik et al. [52]. This diffractogram is also like those from the ASTM 9-453 file [48], or what was obtained by Brindley and Robinson [53]. All peaks determined in our diffractograms can be found in the diffraction patterns of Szczepanik et al. [52] including the one at d = 3.806 Å, which is missing from the publication of Brindley and Robinson [53].

Kaolinite has been identified as an impurity in the bentonite raw material. It was identified in some XRD analyzes by the peak at d = 7.14 Å (Figure 9). The presence of this mineral is certain but the quantities in which it appears are very small. The very sharp peak indicates a well crystallized kaolinite, but the low intensity indicates its low amounts.

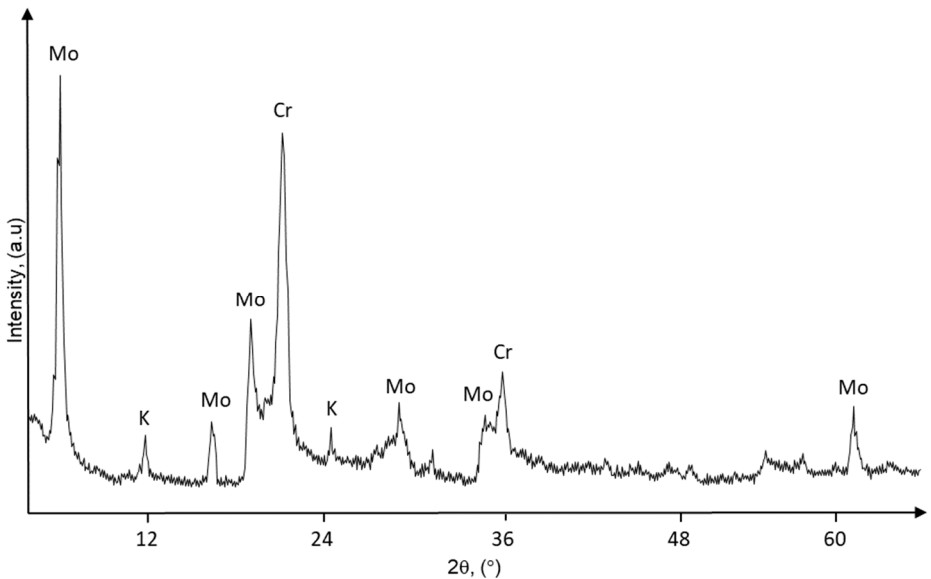

**Figure 9.** XRD of a bentonite sample containing montmorillonite and cristobalite with small amounts of kaolinite (Mo—Montmorillonite; Cr—Cristobalite; K—kaolinite).

Quartz is present in bentonites, especially in the sandy fraction (Table 2), while in the fine fraction its proportion is only 1–2%. It was identified by XRD analyzes and can be observed in Figures 7 and 8.

Feldspars are present in the sandy fraction (Table 2) and have been identified by XRD analysis (Figures 7 and 8) in association with cristobalite and quartz. Its proportion can reach up to 10%

### 4.4. Thermal Analysis

The results of the thermogravimetric analysis of 600 mg nano-/microscale sample can be seen in Figure 10a.

The strong endothermic effect $P_1$ from the TG graph displays a mass loss of 10.66% equal to 64 mg. It then can be observed in the DTG between 30–220 °C, with a maximum around 120–130 °C, and is well correlated with the DTA curve between 30–240 °C with a maximum around 140–150 °C. Its presence is explained by the dehydration of absorbed water suffered by the montmorillonite in this temperature domain [49,54]. The peaks are symmetrical, sharp and cover a large area inside them. A superimposed peak can also clearly be observed with a maximum at 180–190 °C in the DTG curve and between 195–210 °C on the DTA. This is a characteristic to Cheto-type Ca-montmorillonites and is explained by dehydration of Ca-bound water.

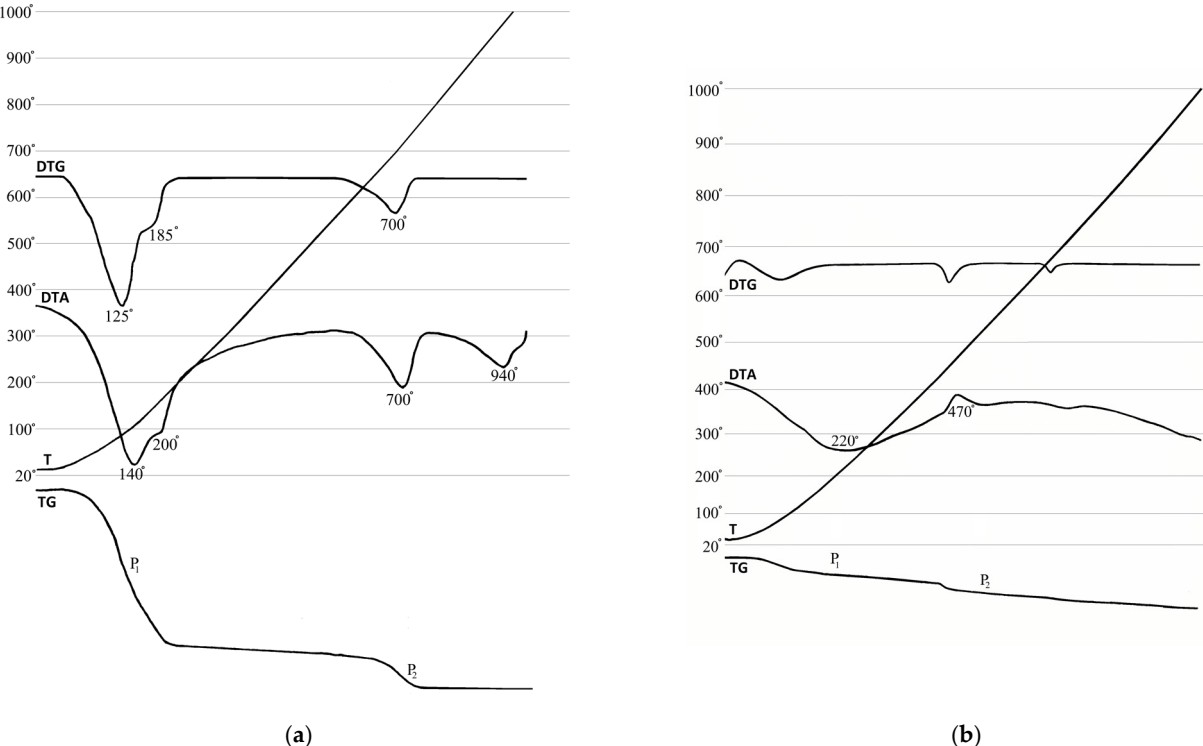

**Figure 10.** Thermal analysis of the montmorillonite and cristobalite: DTG—derivative thermogravimetry; DTA—differential thermal analysis; TG—thermogravimetry; T—temperature 20–1000 °C, $P_1$ and $P_2$—weight losses. (**a**) white nano-/microstructured Orașu Nou-montmorillonite; (**b**) cristobalite.

The second endothermic effect $P_2$ from the TG curve displays a mass loss of 25 mg representing 4.16%. It can be observed between 700–720 °C on DTG and 700–740 °C on the DTA curves. It is explained by the elimination of the hydroxyl groups from the octahedral positions of the crystalline structure.

The DTA curve displays two more thermal effects, but without mass loss: a weak endothermic one between 930–940 °C, followed by an exothermic at 980 °C. This region's "S"-shaped curve characterizes the Wyoming-type Na-montmorillonites. The endothermic effect is due to the transition of the mineral's ordered crystal structure to a disordered system. The second, exothermic one shows an internal reorganization of the components that resulted from burning the mineral, leading to a lower energy state. These results can be compared to those from other studies [2,49].

Comparing the XRD and thermal analyses seems to lead to contradictory results, some of them indicating the presence of Ca-montmorillonite while others being characteristic to Wyoming-type Na-montmorillonites [49,54]. However, the conclusion of these analyses suggests the presence in the Orașu Nou sample of mixed Ca-montmorillonite with some Na-type.

The thermal behavior of cristobalite (Figure 10b) was studied on the same sample investigated by X-ray diffraction. The maximum endothermic effect occurs at a temperature of 220–230 °C. The bell-shaped effect is caused by the transformation of α-cristobalite into β-cristobalite [55]. No thermal effect appears on the DTG curve in this range, while, on the TG curve, there is a slight loss of about 1.125%, due to the removal of adsorbed water in cristobalite.

### 4.5. FT-IR Spectra of Montmorillonite

The FTIR transmission spectra of the nano-/microstructured material can be seen in Figure 11, while a comparative summary with bibliographic montmorillonite data is presented in Table 5.

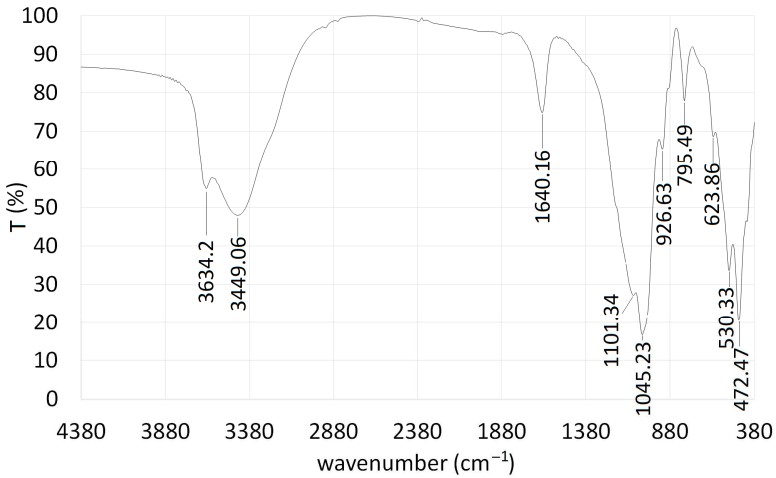

**Figure 11.** FTIR transmission spectra for the white nano-/microstructured Orașu Nou-montmorillonite separate.

**Table 5.** Comparative FTIR bands of the Orașu Nou bentonite with bentonites from scientific data base.

| Castellini et al., 2017 [41] | Krupskaya et al., 2017 [47] | Oinuma and Hayashi 1965 [56] | Tyagi et al., 2006 [57] | Orașu Nou sample (cm$^{-1}$) | Interpretation |
|---|---|---|---|---|---|
| 467 | 470 | 470 | | 472 | Si-O-Si bending in tetrahedra |
| 523 | 523 | 520 | 529 | 530 | Si-O-Al bending in tetrahedra |
| 628 | | 625 | | 624 | Si-O-Si from cristobalite |
| | | 694 | 692 | 694 | Si-O-Si from quartz |
| 793 | | 795 | 793 | 795.5 | Mg-OH-Fe bend |
| 886 | 876 | 835 | 875 | 891 | Al-OH-Fe bending |
| 916 | 925 | 912 | 915 | 926 | Al-OH-Al bending |
| 1042 | 1053 | 1020 | 1035 | 1045 | in plane Si(Al)-O stretching |
| 1086 | 1095 | 1130 | 1113 | 1101 | Si-O out of plane stretching |
| 1640 | 1635 | 1640 | 1639 | 1640 | Absorbed water bending |
| 3434 | 3429 | 3406 | 3440 | 3449 | Absorbed water stretching |
| 3630 | 3634 | 3635 | 3623 | 3634 | octahedra stretching with OH corners linked to Al or Mg |

One can see a very good correlation between our measured data and the ones from previously published studies [47,49,56]. The value from 795.5 cm$^{-1}$ does not correspond to montmorillonite-related data from the literature, instead it was previously attributed to cristobalite or to disordered tridymite [56]. This idea holds especially because these minerals were identified in the XRD patterns. The band at 692 cm$^{-1}$ shows the presence of microcrystalline quartz, while the vibration band from 794 cm$^{-1}$ was previously attributed to illite, hydrated illite or interstratified illite-montmorillonite [49]. Because no interlayered illite-montmorillonite can be seen in the XRD pattern, this vibration is probably due to small amounts of illite, identified in the XRD patterns of the bulk.

The supplementary band indicated in the literature at 426 cm$^{-1}$, corresponding to Si-O vibrations [47], is not shown in our table because it cannot be seen in our FTIR spectra. This band was attributed to the presence of kaolinite [58], so that its absence from our measurements indicates the absence of this mineral from the sample.

The measured transmission spectrum is dominated by the Si(Al)-O in-plane stretching of montmorillonite, around 1045 cm$^{-1}$, close to values from the literature [41,57]. The same authors also identify the 1101 cm$^{-1}$ band for out-of-plane Si-O stretching. The 472 cm$^{-1}$ bending is a general feature exhibited by all clay minerals heaving this bond. Considering together the analysis of the 926 cm$^{-1}$ vibration of the Al-Al-OH hydroxyl groups and the one with iron substitute at 891 cm$^{-1}$ for Al-Fe-OH confirms this substitution in the octahedral groups [59]. This Al-Al-OH vibration is not considered to be characteristic

only to montmorillonite, but also to kaolinite [57,59]. In our case it can be associated to montmorillonite because the XRD pattern of this nano-/microscale fraction does not show the presence of kaolinite. However, this vibration can probably be partially attributed to halloysite also, which was identified in the XRD patterns and in the SEM micrographs of the raw white bentonite sample.

The surface absorbed water bending and stretching vibrations from 1640 cm$^{-1}$ and 3449 cm$^{-1}$, are wide and deep because Ca-montmorillonite retains high quantities of water, even after the mentioned overnight desiccation [60].

The band at 3634 cm$^{-1}$ is attributed to stretching of the OH from octahedral positions when it is linked to Al$^{3+}$ or its bivalent substitutes [41,57].

The general picture of the FTIR spectroscopy leads to the conclusion that the Orașu Nou montmorillonite's nano-/microstructured part is manly of Ca-type.

### 4.6. Chemical Composition

The complete silicate analysis is presented in Table 6. The bentonite has a high Al$_2$O$_3$ content due to high amounts of montmorillonite. Higher amounts of CaO than that of Na$_2$O indicate that we mainly deal with a Ca-type montmorillonite, which is in perfect accordance with the thermal analysis results. High amounts of interstitial H$_2$O$^-$, especially in the last two samples, indicate the presence of large amounts of montmorillonite in the fine fraction of the bentonite samples.

**Table 6.** Chemical composition of bentonite (1—bentonitized perlite; 2—bentonite; 3—bentonite fraction +0.01; 4—bentonite fraction −0.01).

| Sample Codename | SiO$_2$ | Al$_2$O$_3$ | Fe$_2$O$_3$ | MgO | CaO | Na$_2$O | K$_2$O | TiO$_2$ | P$_2$O$_5$ | CO$_2$ | S | H$_2$O$^+$ | H$_2$O$^-$ |
|---|---|---|---|---|---|---|---|---|---|---|---|---|---|
| 1 | 59.10 | 20.30 | 1.43 | 1.50 | 1.54 | 0.86 | 1.00 | 0.25 | 0.04 | 3.90 | 0.51 | 2.17 | 7.88 |
| 2 | 61.50 | 18.15 | 1.70 | 1.55 | 1.55 | 0.33 | 0.66 | 0.17 | 0.04 | 3.84 | 0.12 | 2.00 | 8.82 |
| 3 | 57.21 | 18.95 | 2.00 | 1.95 | 1.75 | 0.10 | 0.65 | 0.16 | 0.03 | 3.95 | 0.20 | 2.60 | 10.50 |
| 4 | 52.50 | 22.85 | 1.50 | 1.95 | 1.90 | 0.11 | 1.32 | 0.15 | 0.02 | 4.20 | 0.27 | 3.12 | 10.03 |

The SEM-EDXS results (Table 7) can be used only as a guideline for major element composition determinations in the prepared nano-/microscale separate [40]. This is caused by the mixing of quite large quantities of cristobalite with the montmorillonite, and the probable presence of a very low content, below the quantifying power of previous determinations, of halloysite fragments, kaolinite, illite, and of some iron oxyhydroxides. Therefore, the mass concentration of Si can originate from many minerals. However, high Ca and very low Na amounts are clearly determinable. Some Fe and lower Mg contents substitute the Al and K, while other substitutes can also be detected. This determination also suggests a mainly Ca-type montmorillonite.

**Table 7.** Major elements determined by SEM-EDXS.

| Element | wt (%) | $\sigma_{Wt}$ (%) |
|---|---|---|
| Si | 27.97 | 0.2 |
| Al | 9.68 | 0.09 |
| Fe | 1.79 | 0.1 |
| Ca | 1.21 | 0.05 |
| Mg | 0.38 | 0.03 |
| K | 0.19 | 0.04 |
| Ti | 0.11 | 0.05 |
| Mn | 0.04 | 0.06 |
| Na | 0.03 | 0.03 |

The pH measurements average 6.2, slightly below the neutral value, covering the 5.0–7.2 interval. Lower values appear for the poorly bentonitized rocks with high quantities of quartz and cristobalite.

One of the fundamental properties of bentonites is their high cation exchange capacity (CEC). The main exchangeable cation contents, for $Na^+$, $K^+$, $Mg^{2+}$ and $Ca^{2+}$ are presents in Table 8. The anion exchange capacity increases with decreasing pH. Highest CEC values were obtained for $Ca^{2+}$, followed, in this order, by $Mg^{2+}$, $Na^+$ and $K^+$. It can also be seen that the highest CEC values are like those of the typical smectite-rich soil. These values are a little bit higher than those reported for other similar minerals [2,44].

**Table 8.** Physical properties of the studied bentonites and bentonitized rocks.

| Physical Properties | Maximum | Minimum | Average |
| --- | --- | --- | --- |
| pH | 7.20 | 5.00 | 6.2 |
| CEC-$Ca^{2+}$ (cmol/kg) | 70.79 | 16.86 | 45.89 |
| CEC-$Mg^{2+}$ (cmol/kg) | 28.31 | 0.99 | 9.5 |
| CEC-$K^+$ (cmol/kg) | 28.66 | 0.85 | 5.14 |
| CEC-$Na^+$ (cmol/kg) | 65.96 | 1.24 | 5.29 |
| Specific weight (g/cm$^3$) | 2.73 | 2.06 | 2.28 |
| Absorption capacity (ml/g) | 66.12 | 27.84 | 43.58 |
| Degree of swelling (%) | 11.40 | 3.40 | 9.41 |
| Plasticity index (%) | 134.80 | 41.25 | 68.87 |

CEC measurements correlate well with the major element composition from Table 8. The elemental concentration variation is reflected in CEC with the notable exception of $Na^+$. Although it represents a low mass percent it has a relatively high CEC.

Low specific weight values indicate the abundance of clay minerals. High absorption capacity values result in a high mean swelling capacity of around 9.41 times more than the volume of the dry bentonite. Adsorption capacity is correlated with morphology and surface exchange, elemental composition, and crystallinity of the raw bentonite materials, [61]. High swelling capacities also induce high plasticity values, which fall in the domain of very high plasticity.

*4.7. Summary Discussion*

The presence of iron oxyhydroxides can be identified with the bare eye, from the rusty-looking infiltrations in the cracks of the raw mineral. They concentrate in the fine fractions during separation, giving a weakly reddish color. Their presence could not be determined by XRD in our nano-/microscale separate, because the sample was purposely chosen in such a way that it visually exhibited as small amounts as possible of such contamination. However, their presence cannot also be excluded and could be correlated to the high Fe contents found from the SEM-EDX measurements, beside the Fe substitute in montmorillonite [62].

Mineralogical and chemical data of the Oraşu Nou bentonite suggest that the alteration process to bentonite was mainly a hydrothermal-deuteric one. Deuteric weathering affected the pyroclastic materials and the perlites in aqueous media with an alkaline pH, with high Mg and Ca concentrations, but lacking K [15]. From a mineralogical point of view, the chemical transformations occur mainly as formation of clay minerals, montmorillonite being the prevailing one accompanied by cristobalite [33].

The mineralogy of bentonite and especially of the opal phases are controlled mainly by the regional geology and the composition of the host rocks. Opaline silica from bentonite is hydrated because it is formed at low temperatures and provides an indication on bentonite formation. The Oraşu Nou bentonites, with high cristobalite content, can be considered products of both hydrothermal and diagenetic origins [63].

Watanabe et al. [64] proposed the hypothesis that halloysite could be formed by the degradation of montmorillonite in the environment simultaneously with the precipitation

of cristobalite. This process may also have occurred in the Oraşu Nou bentonites, because halloysite is associated with cristobalite.

The Oraşu Nou Bentonites are important industrial raw materials because of their main component, montmorillonite, which gives them a very high sorbent property and swelling capacity.

This study of the Oraşu Nou bentonite reveals a good quality raw material that can be used in many fields of activity: agriculture [22] to immobilize fungicides or insecticides [25], inactivate heavy metals in polluted soils, especially zinc [28], nuclear waste storages [32,33].

The pregnant presence of cristobalite may have an impact on the potential use of the Oraşu Nou bentonites only in certain areas and especially in medicine [9]. Because no cristobalite was identified in the nano-/microscale bentonite separate, this material might also be used in medical applications.

## 5. Conclusions

This is a general study using physico-chemical methods on the mineralogical composition and possible applications according to the determined physico-chemical properties of the bentonitized rocks and the white bentonite along its nano-/microstructured separate prepared by us, from the deposit near Oraşu-Nou (Romania). The main mineral found in this bentonite deposit is montmorillonite. Beside it, small quantities of kaolinite, halloysite, illite, cristobalite, quartz, carbonates, and iron oxyhydroxides appear. This study revealed that the deposit formed after a hydrothermal-deuteric alteration of ignimbrite and pyroclastic volcanic rocks. The bentonitic material's average pH is just below neutral, around 6.2, and its cation exchange capacity is averaged to be 45.89 cmol/kg. XRD, FT-IR and thermogravimetric studies of the hand-milled sample give characteristic results for montmorillonite, leading to a mainly Ca-type one and subordinately a Na-type. SEM imaging of the raw mineral proves the presence of lamellar montmorillonite and indicates possible small quantities of halloysite and other silica polymorphs. Halloysite was determined using XRD and from the SEM micrographs as nano-/microscale rod-like structures. Low temperature cristobalite can be identified in most of our XRD patterns, while its presence was conformed using thermogravimetric analysis. This mineral can be found in large quantities in the coarse fraction, mainly in the poorly bentonitized rocks, where lower montmorillonite concentrations were also determined. In relatively lower percentages it can be also found in the bentonites, but it is also present in low quantities in the finest fractions.

This work proves the presence of a high quality, easily and inexpensively refinable, high Ca-montmorillonite content bentonite deposit, especially when we are talking about the white-colored raw material. Because the immediate neighborhood of this deposit is also a region rich in this mineral, which, moreover, can be exploited using open-pit mining, real opportunities are opened for a feasible harness of this resource and its use in many industrial branches, agriculture, environmental protection, or in modern applications of nano-/microscale materials.

**Author Contributions:** Conceptualization G.D.; methodology G.D., Z.S., G.I., D.A.; formal analysis G.D., F.D., Z.S., G.I., D.A.; investigation G.D., F.D., Z.S., G.I., D.A.; resources G.D., F.D., Z.S., G.I.; data curation, G.D., F.D., Z.S., G.I., D.A.; writing—original draft preparation, G.D., F.D., Z.S.; writing—review and editing G.D., F.D., Z.S.; visualization Z.S., G.I.; supervision, G.D.; project administration, Z.S.; funding acquisition G.D., Z.S. All authors have read and agreed to the published version of the manuscript.

**Funding:** The authors acknowledge the financial support of JINR Dubna-TU Cluj-Napoca Joint Research Projects.

**Acknowledgments:** The authors acknowledge that the FT-IR analyses were made at Faculty of Chemistry, the XRD analyses were made at the Faculty of Physics from the "Alexandru Ioan Cuza" University, Iaşi, Romania, SEM imaging and EDXS analyses at the Geological Institute of Romania.

**Conflicts of Interest:** The authors declare no conflict of interest. The funders had no role in the design of the study; in the collection, analyses, or interpretation of data; in the writing of the manuscript, or in the decision to publish the results.

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
