# Peer review of "Mineralogical and Physico-Chemical Characterization of the Oraşu-Nou (Romania) Bentonite Resources"

_minerals, doi:10.3390/min11090938_

Round 1

Reviewer 1 Report

Manuscript ID- minerals-1350594

Title: Mineralogical and physico-chemical characterization of the Oraşu-Nou (Romania) bentonite resources

Authors: Gheorghe Damian, Floarea Damian Zsolt Szakács, Gheoghe Iepure and Dan Aştefanei

The title of the paper is interesting as it focuses on a local bentonite; this study is interesting for local industries. The presentation of the paper is adequate.

This paper could be proposed for publication if the authors improve the manuscript according to the indications made by the reviewer.

However, several comments are made to the authors below:

Abastract should be reorganised

The authors have used several very old references. There are many recent works on the subject of bentonites.

Most of the images are of poor quality

The authors have not considered some very necessary tests for the correct characterisation of bentonites, such as the Oriented Aggregate Analysis. This type of analysis is important to analyse samples of complex mineralogical and petrological constitution, as is the case of the bentonites investigated in this work.

Abstract. Line 18: write "fraction" instead of "fration".

Abstract: the authors have to write the objective pursued with this research and in which field they want to apply it. Rewrite.

Introduction: the field of bentonites is very broad and well studied. The authors should expand the description and citation of published works. Por ejemplo: https://doi.org/10.3390/cryst11060706; https://doi.org/10.1016/j.porgcoat.2017.05.018; https://doi.org/10.3390/min9110696; https://doi.org/10.1139/cgj-2020-0045; doi.org/10.1016/j.clay.2021.106024; doi.org/10.1016/j.jafrearsci.2021.104258; doi.org/10.1016/j.clay.2021.106049.

Section 2. Geological setting: Lines 63 to 69: I consider that the reference is missing. Add

Section 2. Materials and methods: I recommend dividing into two subsections. 2.1: Materials; 2.2: Methods.

Subsection 3.6: Write "Chemical composition" instead of "Physico-chemical properties".

Line 70. Figure 1: The quality of the image should be improved and the reference should be added. Revise the quality of all figures in the manuscript

Lines 86 to 94: Samples should be correctly identified by name or by number; furthermore, their specific characteristics should be studied in depth; weight in natural state among others. It is recommended to include a map of the location of the sampling points within the deposit (use the Google Earth database and reference it).

Lines 95 and 96. Figure 2: Image quality is very poor. It needs to be improved

Line 140. Subsection 3.1: Delete the word "Synthetic".

Lines 170 to 171: Indicate the mineral phases within the micrograph. The SEM discussion is very poor. Should go into more depth.

Lines 180 to 181. Figure 4: In the XRD patterns not all the mineralogical phases mentioned in Table 2 are identified. In addition, the discussion is poor. Rewrite.

Lines 280 to 281: Write "Al2O3" instead of "Al"; write "CaO" instead of "Ca"; write "Na2O" instead of "Na".

Line 295: Write "Al," instead of "Al."

Section 4. Lines 357 to 371: The determination of the Genesis is not part of the objectives of this research, nor is it reflected in the Abstract. Therefore, the data discussed in this section seem to belong to other authors. This Section has to be deleted.

Lines 373 to 377: This paragraph should be moved to Conclusions.

Lines 378 to 399: These paragraphs should be moved to the Introduction Section. Section 5 is of no interest, it should be deleted.

All images need to be enhanced

Reviewer 2 Report

Thank you for the manuscript "Mineralogical and physico-chemical characterization of the Oraşu-Nou (Romania) bentonite resources"

I have read through the manuscript and found that overall, the manuscript has to go through major revision in order to be considered for publication. 

My main concerns are related to the following points:

  • Introduction is weak and messy. Sections 4 and 5 comes late in the manuscript but should be part of the introduction. Also, section 2 "geological setting" should be included as part of the "Introduction". Most importantly, the text under "geological setting" lacks references and must be updated to reach the required standard for publication. Please rewrite and include references to the geolgical knowledge.
  • In the introduction also add some info about cristobalite (and tridymite) in naturally occuring rocks, how these minerals are deposited and how they are detected in the rock samples and challenges related to this.
  • Section 2 "Materials and Methods" as it is in the manuscript Section 2 comes twice (geological setting + materials and methods). Update! Further, a lot of the text in the "materials and methods" line 85 to 92 is motivation of study. Please include this in the "Introduction" and rewrite to include a precise statement of objective and goals for the study.
  • Further, "materials and methods" must be revised to include a more structured and correct presentation of analytical methods and setting. E.g. SEM imaging and EDX seem to have been performed on the same SEM equipment. Please make a complete description of the SEM equipment. Also provide more specific info about LOI analyses. What temperature and for how long?
  • Section 3 "Results" this must either be called "Results and discussions" or rewritten to 3 "Results" and 4"Discussion". The latter is to preferr for readability.
  • Section 3 "Results" how was Cristobalite confirmed in the samples? It is well known that other minerals, e.g. Opal C, easily can be mistaken for Cristobalite. Please provide more details. And why have the authors not used the DTA spectra to confirm and quantify Cristobalite? alpha cristobalite has a distinctive reversible transformation to beta-cristobalite between 200-275 deg C. Can be used as support.
  • In "Materials and methods" authors mention "representative sample" - what is the sample representing? A representative sample commonly has a well thought sampling procedure in the base in order to represent a certani volume, geological unit, ore body section etc. Please specify in which way samples are representative!
  • Picture in Figure 2 is low quality. Please provide picture with better quality (resolution and details of features in the rock).
  • Figure 3 - please make annotations to the mentioned structures on the micrograph to explain "lamellar" structure of montmorillonite and the rod-like structure of halloysite.
  • In the presentation of the results, tables and graphs need to be updated with reference to which analytical method the data presented have been produced from (e.g. TGA, ICP ...).
  • In general, the structure and presentation of the manuscript needs serious improvement.

These are my main concern in this round and needs to be improved before new evaluation. 

Reviewer 3 Report

In this manuscript the authors describe a Neogene bentonite clay using physico-chemical methods. Their goal is to go beyond the conventional applications of commercial bentonite and demonstrate how very high-quality colloidal suspensions can be obtained which can be used in the most modern applications of micro- and nanostructured materials. Using first rate analytical methods their results include the mineralogical composition identified using X-ray diffraction, Fourier transform infrared spectroscopy, thermogravimetric analysis, scanning electron microscopy coupled with energy dispersive X-ray spectroscopy, pH and cation exchange capacity determinations for different bentonites and associated pyroclastic rocks. The average mass fraction of montmorillonite is between 35% and 75%, reaching up to 95% in certain samples in the deposit. Montmorillonite is mainly calcic, and subordinately sodic. A small amount of halloysite and very fine cristobalite were also identified in the fine faction. The natural bentonite’s average pH is 6.2 and its cation exchange capacity is in the lower-mid range for smectites. Their study concluded that that the deposit formed after a hydrothermal – low-temperature magmatic alteration of ignimbrite and pyroclastic volcanic rocks. And it demonstrates the presence of a high quality, easily and inexpensively refined, high Ca-montmorillonite content bentonite deposit.

            The authors have done an excellent job of describing and analyzing the bentonite samples from this part of Romania. Their results will most certainly be helpful to scientific and industrial applications of nanoparticle science. I have only a couple of small suggestions for the authors that will help clarify their presentation. First, their X-ray diffraction data is quite helpful, but would be even more revealing if they would include ethylene glycol saturated and heated diffraction patterns. For the average reader these data should clarify the identification of the various clay mineral species. And I notice one small typo in line 268. It should read, “..does not show the…” Beyond that, the figures and the references are excellent and I’m sure this report will be of interest to a wide range of specialists.

Round 2

Reviewer 1 Report

The authors have improved the manuscript according to the reviewer's indications. The manuscript is now of a better quality in all respects.

The reviewer recommends the authors to make some minor changes, as indicated below:

In Figure 2 you have to put the Google Earth reference number "[X]".
You have to put the Google Earth citation in the References Section and add the date of access to Google Earth. You can use the following example for the citation: [68] Google Earth. Available online: https://earth.google.com/web/@36.77840096,-2.06997867,56.72506538a,276.38480656d,35y,0h,0t,0r (accessed on 15 July 2021).

It is recommended to approve the manuscript as soon as the authors make the minor changes indicated by the reviewer.

Author Response

Response to Reviewer 1 Comments round 2

The reviewer recommends the authors to make some minor changes, as indicated below:

Point 1: In Figure 2 you have to put the Google Earth reference number "[X]".
You have to put the Google Earth citation in the References Section and add the date of access to Google Earth. You can use the following example for the citation: [68] Google Earth. Available online: https://earth.google.com/web/@36.77840096,-2.06997867,56.72506538a,276.38480656d,35y,0h,0t,0r (accessed on 15 July 2021).

Response 1: We have introduced the citation in the text and references according to your request.

Reviewer 2 Report

Dear authors,

I appreciate the quick and thorough revision of the manuscript.

I will still suggest a english language check to make the language more flowing and to avoid possibility to misunderstand, do to the language.

I will also suggest to slightly restructure the "Results and discussio", where headings are very method specific in 3.2 to 3.6 but 3.7 then repeats all this for Cristobalite and Hyalite specifically. Would be better to include also these in 3.2 to 3.6. Then 3.7 can be a more summary discussion(?)

The introdution is improved but still lack some presentation of the background for some of the metodology and specifically about Cristobalite properties.
